# IgBlend: Unifying 3D Structures and Sequences in Antibody Language Models

## Abstract

Large language models (LLMs) trained on antibody sequences have shown significant potential in the rapidly advancing field of machine learning-assisted antibody engineering and drug discovery. However, current state-of-the-art antibody LLMs often overlook structural information, which could enable the model to more effectively learn the functional properties of antibodies by providing richer, more informative data. In response to this limitation, we introduce IgBlend, which integrates both the 3D coordinates of backbone atoms (C-alpha, N, and C) and antibody sequences. Our model is trained on a diverse dataset containing over 4 million unique structures and more than 200 million unique sequences, including heavy and light chains as well as nanobodies. We rigorously evaluate IgBlend using established benchmarks such as sequence recovery, complementarity-determining region (CDR) editing and inverse folding and demonstrate that IgBlend consistently outperforms current state-of-the-art models across all benchmarks. Furthermore, experimental validation shows that the model's log probabilities correlate well with measured binding affinities.

## 1 Introduction

Antibodies are key components of the adaptive immune system, capable of recognizing and neutralizing a wide range of pathogens, including viruses, bacteria, and other foreign invaders. Their ability to bind specific targets with high affinity makes them essential tools in therapeutic development. Recent advancements in natural language processing (NLP) have led to the creation of foundational language models that can learn from and modify antibody sequences (Olsen et al., 2022b; 2024; Prihoda et al., 2022). Moreover, the three-dimensional (3D) structure of an antibody is closely linked to its specificity, affinity, and interaction with antigens. Therefore, capturing the relationship between sequence and structure is crucial for tasks such as affinity maturation, de novo antibody design, and optimizing antibody-antigen interactions for therapeutic applications. While current language models excel at either sequence-to-sequence or structure-to-sequence (inverse folding) tasks, relying on only one of these modalities at the input limits their capability and flexibility in more complex antibody engineering tasks (Olsen et al., 2022b; 2024; Prihoda et al., 2022; Høie et al., 2023). In this paper, we introduce IgBlend, a multi-modal model designed to incorporate both sequence and structural information for antibody engineering. Our approach can utilize either sequence, structure, or both, enabling the model to not only sample sequences that can fold to the same parental backbone but also generate more diverse sequences, providing greater flexibility in designing antibody sequences. Moreover, by utilizing both experimentally resolved structures (Dunbar et al., 2014) and synthetic data generated through structure prediction models (Abanades et al., 2023b; Ruffolo et al., 2023), we aim to improve model performance on key antibody engineering tasks. Our contributions can be summarized as follows:

- We introduce IgBlend, a model that learns antibody representations from either sequence, structure or sequence-structure pairs when structural data is available.

- We present a pre-training strategy with multiple sub-objectives as well as a procedure for training and dataset processing, all of which can broadly be applied to other multi-modal training settings.

- We empirically demonstrate that integrating structural information, even when synthetically generated, significantly improves the performance of large models across a wide range of benchmarks.

The remainder of the paper is organized as follows. First, we review related works and introduce the notations. Section 2 details the architecture of IgBlend, datasets, and training procedures. Then, we compare the performance of IgBlend against existing models in Section 3. Finally, we point out that a more detailed background on antibodies can be found in Appendix A.

**Related work.** In recent years, significant progress has been made in developing protein and antibody foundation models, drawing from advances in natural language processing (NLP). These models can be broadly categorized based on their focus—either on general protein design or antibody-specific tasks—and the way they approach the problem, such as sequence-to-structure prediction, structure-to-sequence generation, or sequence-structure co-design. For general protein design, sequence-to-structure models, including AlphaFold (Jumper et al., 2021) and RoseTTAFold (Baek et al., 2021), have significantly improved the prediction of protein structures from amino acid sequences. In the structure-to-sequence domain (inverse folding), ESM-IF (Hsu et al., 2022) predicts amino acid sequences that fold into a given structure. Meanwhile, sequence-to-sequence models, such as ESM (Rives et al., 2021) and its variants, excel at identifying patterns within sequences for tasks such as sequence recovery and mutation prediction. Moreover, some recent works have focused on sequence-structure co-design, with models such as ESM3 (Hayes et al., 2024) incorporating both sequence and structure as well as function to improve protein design. There is also hybrid approaches such as LM-Design (Zheng et al., 2023) that leverage both sequence and structural inputs to design new protein sequences, aligning with the approach used in this paper.

In the context of antibodies, several models have emerged with a similar framework but tailored for immunoglobulins. For sequence-to-structure prediction, antibody-specific models such as ImmuneBuilder (Abanades et al., 2023b) and IgFold (Ruffolo et al., 2023) predict 3D antibody structures from sequence data. In the structure-to-sequence domain, AntiFold (Høie et al., 2023) addresses the inverse folding problem, predicting antibody sequences that correspond to a given backbone structure. For sequence-to-sequence tasks, models such as AbLang (Olsen et al., 2022b), AbLang-2 (Olsen et al., 2024), AntiBERTy (Ruffolo et al., 2021), and Sapiens (Prihoda et al., 2022), which are predominantly based on the BERT architecture (Devlin et al., 2018), are specifically designed for antibody sequences, helping to improve performance on tasks such as residue restoration and paratope identification. To the best of our knowledge, although these models focus on either sequence or structure, no existing antibody-specific LLM effectively integrates both modalities. In this paper, we address this gap by introducing a sequence-structure-to-sequence framework, similar to LM-Design (Zheng et al., 2023) for proteins, to learn a richer and more informative representation. IgBlend learns joint representations of structure and sequence during pre-training, improving upon models that rely on a single modality. Finally, we leave out diffusion, flow matching, and graph-based approaches to antibody design to maintain our focus on language models.

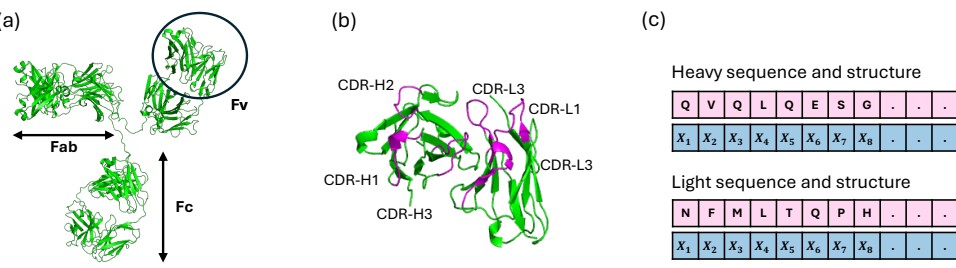

Figure 1: (a) Antibody structure with antigen binding (Fab), crystallizable (Fc), and variable (Fv) regions, (b) Zoom over the variable region which contains an heavy and a light chain, CDRs regions are displayed in magenta, (C) Modalities that we exploit in this paper for antibody modeling.

**Notations.** For any single unpaired chain (heavy, light or nanobody), we denote the backbone structure and sequence of the chain with $n$ residues as follows:

$$\text{structure: } \mathbf{x} := (\mathbf{x}_1, \ldots, \mathbf{x}_n) \in \mathbb{R}^{3 \times 3 \times n} \text{ and sequence: } \mathbf{s} := (\mathbf{s}_1, \ldots, \mathbf{s}_n) \in \mathbb{A}^n$$

where $\mathbf{x}_i \in \mathbb{R}^{3 \times 3}$ represents the 3D coordinates of the C-alpha, N, and C atoms of the $i^{th}$ residue, while $\mathbf{s}_i \in \mathbb{A} := [A, R, N, D, C, E, Q, G, H, I, L, K, M, F, P, S, T, W, Y, V, *]$ specifies the amino acid type corresponding to the $i^{th}$ residue, where $i \in \{1, \ldots, n\}$. For consistency in notation, we will also use $*$ to denote the unknown token for both structure and sequence tokens, acknowledging a slight abuse of notation. Moreover, we stress that this work solely focuses on unpaired sequences and leaves the fine-tuning on purely paired sequences for future work. In the remainder of this paper, we will use $\mathbb{P}$, $\mathbb{E}$, and $\mathbb{I}$ to denote the standard probability, expectation, and indicator functions, with $\mathbb{I}$ specifically taking values in $\{0, 1\}$. To compute the differences between two sequences $(\mathbf{s}, \widehat{\mathbf{s}}) \in \mathbb{A}^{|\mathbf{s}|}$ of the same length, we will use the normalized Levenshtein distance: $\text{Levenshtein}(\mathbf{s}, \widehat{\mathbf{s}}) = (1/|\mathbf{s}|) \cdot \sum_{i=1}^{|\mathbf{s}|} \mathbb{I}\{\mathbf{s}_i \neq \widehat{\mathbf{s}}_i\}$. To compute differences between two backbone structures $(\mathbf{x}, \widehat{\mathbf{x}}) \in \mathbb{R}^{3 \times 3 \times |\mathbf{x}|}$, we will use the Root Mean Square Deviation (RMSD) defined as $\text{RMSD}(\mathbf{x}, \widehat{\mathbf{x}}) = \arg\min_{R \in \Omega_3, t \in \mathbb{R}^3} (1/3|\mathbf{x}|) \cdot \sum_{i \leq |\mathbf{x}|, j \leq 3} \|\mathbf{x}_i^j - R^* \widehat{\mathbf{x}}_i^j - t^*\|_2^2)^{1/2}$ where $R^* \in \mathbb{R}^{3 \times 3}$ and $t^* \in \mathbb{R}^3$ respectively denote the optimal rotation matrix and translation after finding the optimal rigid alignment with the Kabsch algorithm (Kabsch, 1976) between the backbone structures where $\Omega_3 \subset \mathbb{R}^{3 \times 3}$ denotes the set of 3D rotations and $\|\cdot\|_2$ denotes the standard Euclidean distance.

## 2 METHODS

In this section, we present the IgBlend architecture, the dataset, and the pre-training objectives.

### 2.1 MODEL ARCHITECTURE

The proposed architecture, IgBlend, is illustrated in Fig 2 and consists of three primary components: a structure encoder that handles the backbone coordinates of the antibody, a sequence encoder that processes the amino acid sequence, and a multi-modal trunk that processes both structural and sequential representations.

**Structure encoder.** It generates an abstract representation vector for each set of coordinates $\mathbf{x}_i \in \mathbb{R}^{3 \times 3}$ from the full sequence of coordinates $\mathbf{x} = (\mathbf{x}_1, \ldots, \mathbf{x}_n) \in \mathbb{R}^{3 \times 3 \times n}$. This representation (a 512-dimensional embedding) encapsulates the geometry of the global backbone structure. The architecture comprises four GVP-GNN (Graph Neural Network Geometric Vector Perceptron) layers (Jing et al., 2020), followed by two generic Transformer encoder layers (Vaswani et al., 2017). This design is invariant to rotation and translation of the input coordinates and has been demonstrated to effectively capture protein geometries in various learning tasks (Jing et al., 2020), including

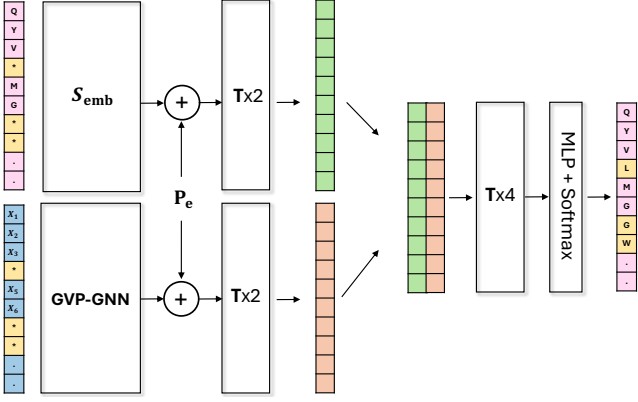

Figure 2: Architecture of the Ig-Blend model. It takes as input both: a series of amino acids (top) and a series of 3D coordinates (bottom). The symbol * denotes either a masked amino acid or a masked set of coordinates. Note that the model can process each modality independently by setting all the tokens of one modality to mask. $S_e$ denotes the sequence embedding (i.e. look-up table), $T$ denotes a transformer block, $P_e$ denotes the sinusoidal position embedding. The sequence encoder is displayed of the bottom left, the structure encoder on the bottom left and on the right the multi-modality processor.

structure-to-sequence models such as ESM-inverse folding (Hsu et al., 2022) and AntiFold (Høie et al., 2023). The input to the encoder consists of a series of residue coordinates $\mathbf{x}$, with a local reference frame established for each amino acid, as per the approach used in AlphaFold2 (Jumper et al., 2021). A change of basis is then performed according to this local reference frame, rotating the vector features from the GVP-GNN outputs into the local reference frame of each amino acid. Finally, the output of the GVP is processed through two Transformer blocks, producing a 512-dimensional embedding for each residue. Notably, each or all sets of coordinates can be masked using the $*$ token.

**Sequence encoder.** In parallel to the structure encoder, the sequence encoder generates a vector representation (i.e., embedding of size 512) for each amino acid $\mathbf{s}_i \in \mathbb{A}$ in the full sequence $\mathbf{s} = (\mathbf{s_1}, \ldots, \mathbf{s_n})$. The architecture includes a one-hot encoded input followed by two blocks of a standard Transformer model (Vaswani et al., 2017). This architecture has already been shown to learn relevant information from antibody sequences (Olsen et al., 2022b; 2024). Specifically, the module utilizes sinusoidal positional embeddings, a SwiGLU activation function (Shazeer, 2020), and has an embedding dimension of 512. Additionally, any amino acid within the sequences can be masked using the masked token $*$.

**Multi-modality encoder.** The fusion layer processes both modalities in two steps. First, it combines the abstract representations from the sequence and structure encoders by concatenating them along the embedding dimension, forming a single vector of size 1024 for each residue. It then processes the concatenated modalities through a series of four Transformer blocks with SwiGLU activations.

**Classification head.** Finally, the classification head consists of a multi-layer perceptron (MLP) followed by a softmax function and processes the multi-modal representation to generate a probability distribution over amino acid types at each position. Further details can be found in Appendix B.

## 2.2 DATA PREPARATION

To create a model capable of processing both sequential and structural information, we compiled (1) a structural dataset $\mathcal{D}_{\text{struct}} = \{(\mathbf{s}, \mathbf{x})_1, \ldots, (\mathbf{s}, \mathbf{x})_{|\mathcal{D}_{\text{struct}}|}\}$, which includes structures paired with their corresponding sequences, and (2) a sequential dataset $\mathcal{D}_{\text{seq}} = \{(s, *)_1, \ldots, (s, *)_{|\mathcal{D}_{\text{seq}}|}\}$, which consists solely of sequence data. These datasets were derived from four sources: SAbDab (Dunbar et al., 2014); PLAbDab (Abanades et al., 2023a); OAS datasets (Olsen et al., 2022a); and INDI (Deszyński et al., 2022) fully described in the Appendix C and summarized in Table 1. The datasets are further divided into samples for heavy chains, light chains, and nanobodies, resulting in $\mathcal{D}_{\text{struct}} = \mathcal{D}_{\text{struct,H}} \cup \mathcal{D}_{\text{struct,L}} \cup \mathcal{D}_{\text{struct,N}}$ and $\mathcal{D}_{\text{seq}} = \mathcal{D}_{\text{seq,H}} \cup \mathcal{D}_{\text{seq,L}} \cup \mathcal{D}_{\text{seq,N}}$. Due to the significant imbalance in the number of samples across modalities, as noted in Table 1, we implemented a new sampling scheme to rebalance the data. For each modality $\text{M} \in \{\text{seq, struct}\}$ and each chain type $\text{C} \in \{\text{L, H, N}\}$, we clustered the datasets $\mathcal{D}_{\text{M,C}}$ using MMseqs2 (Steinegger & Söding, 2017), clustering over the full sequences with the parameters "$-\text{cov-mode 1}$", "$-\text{c 0.8}$", and "$-\text{min\_seq\_id 0.8}$" for the sequential datasets and over the concatenated CDR regions with the parameter "$-\text{min\_seq\_id 0.9}$" for the structure datasets. This process resulted in a union of $n_{\text{cluster}}$ clustered samples $\mathcal{D}_{\text{M,C}} = \bigcup_{i=1}^{n_{\text{cluster}}} \mathcal{C}_{\text{M,C}}(i)$ for each modality and chain type. Based on these clusters, we defined the distributions $\mathcal{P}(\mathcal{D}_{\text{struct}})$ and $\mathcal{P}(\mathcal{D}_{\text{seq}})$ over each dataset modality as follows: first, we sample a chain type C with equal probability: $\mathbb{P}(\text{C} = \text{H}) = \mathbb{P}(\text{C} = \text{L}) = \mathbb{P}(\text{C} = \text{N}) = 1/3$, then we select a sample within the corresponding dataset $\mathcal{D}_{\text{C,M}}$ according to the size of its corresponding cluster:

| Modality | Heavy sequences | Light sequences | Heavy structures | Light structures |
|---|---|---|---|---|
| OAS paired | 1 804 122 | 443 129 | 1 418 312 | 535 130 |
| OAS unpaired | 156 314 998 | 34 464 420 | 1 057 850 | 643 647 |
| PLAbDab paired | 51 740 | 45 620 | 47 554 | 42 021 |
| PLAbDab unpaired | 139 706 | 89 743 | - | - |
| INDI (nanobodies) | 11 231 660 | - | 895 008 | - |
| SAbDab | - | - | 2 056 | 2 024 |
| Total | 169 542 226 | 35 042 912 | 3 420 780 | 1 222 822 |

Table 1: Number of unique samples per modalities and chain types after the first pre-processing step.

$$\mathbb{P}(\mathbf{s}, \mathbf{x})_{|\mathrm{M,C}} = \begin{cases} 1/|\mathcal{C}_{\mathrm{M,C}}(i_s)| & \text{if } (\mathbf{s}, \mathbf{x}) \in \mathcal{D}_{\mathrm{M,C}} \\ 0 & \text{otherwise} \end{cases} \tag{1}$$

where $i_s$ denotes the index of the cluster containing $\mathbf{s}$, and $|\mathcal{C}_{\mathrm{M,C}}(i_s)|$ indicates the size of its corresponding cluster. This clustering-based distribution approach enables us to preserve the entire dataset while re-weighting each cluster to improve diversity in the training set. After clustering, 10% of the clusters are set aside for validation, and another 10% are reserved for testing. Both sets are completely excluded from the training set and have less than 0.8 sequence identity with the training data, ensuring that the validation and test sets are sufficiently distinct from the training set.

## 2.3 PRE-TRAINING OBJECTIVES

To train IgBlend, the data distribution defined by Equation (1) was used, ensuring a balanced representation of heavy, light and nanobodies chains across the two datasets, $\mathcal{D}_{\mathrm{seq}}$ and $\mathcal{D}_{\mathrm{struct}}$. We employ a specialized masked language modeling objective capable of handling both sequential and structural data. The model parameters, $\theta$, are optimized by minimizing the sum of three losses based on cross-entropy:

$$\mathcal{L}_{\text{multi-modal}} := \mathcal{L}_{\text{seq2seq}} + \mathcal{L}_{\text{seq+struct2seq}} + \mathcal{L}_{\text{struct2seq}} \tag{2}$$

where:

$$\begin{cases} \mathcal{L}_{\text{seq2seq}} &= \mathbb{E}_{(\mathbf{s},*)\sim\mathcal{P}(\mathcal{D}_{\text{seq}})} \left[ \sum_{i \in \mathcal{T}_s} -\log(p_\theta(s_i|\mathbf{s}_{/\mathcal{M}_s}, *)) \right] \\[2em] \mathcal{L}_{\text{seq+struct2seq}} &= \mathbb{E}_{(\mathbf{s},\mathbf{x})\sim\mathcal{P}(\mathcal{D}_{\text{struct}})} \left[ \sum_{i \in \mathcal{T}_s} -\log(p_\theta(s_i|\mathbf{s}_{/\mathcal{M}_s}, \mathbf{x}_{/\mathcal{M}_x})) \right] \\[2em] \mathcal{L}_{\text{struct2seq}} &= \mathbb{E}_{(\mathbf{s},\mathbf{x})\sim\mathcal{P}(\mathcal{D}_{\text{struct}})} \left[ \sum_{i \in \mathcal{T}_s} -\log(p_\theta(s_i|*, \mathbf{x}_{/\mathcal{M}_x})) \right] \end{cases} \tag{3}$$

with $p_\theta(\mathbf{s}_i|\mathbf{s}, \mathbf{x})$ denoting the output of the softmax layer shown in Figure 2 at position $i \in \{1, \ldots, n\}$, given $(\mathbf{s}, \mathbf{x})$ as input. By using this combination, the model dedicates equal time on each task. The masking strategy for each pre-training objective is outlined below, defining the positions of the amino acids to predict $\mathcal{T}_s$, the masked residues in the sequence $\mathcal{M}_s$, and the masked structures $\mathcal{M}_x$:

- **seq2seq.** This task, used in training sequence-only antibody models (Devlin et al., 2018; Olsen et al., 2024), is applied to the sequential dataset $\mathcal{D}_{\mathrm{seq}}$, which lacks structural information (i.e., $\mathbf{x} = *$). For each sequence, between 10 and 40 of the amino acids are selected for masking using one of two methods: (i) randomly sampling individual residues throughout the sequence or (ii) masking continuous spans of residues, with the starting position chosen at random. The positions of the residues to be predicted are the same as those masked, $\mathcal{M}_s = \mathcal{T}_s$. The masked residues in $\mathcal{M}_s$ are then processed using one of three strategies: (a) replaced by the unknown token $*$ with 80% probability, (b) substituted with a different amino acid with 10% probability, or (c) left unchanged with 10% probability. The masking distribution is also slightly adjusted to ensure balanced coverage of both CDR and framework regions.

- **seq+struct2seq.** Both sequential and structural information are used to predict masked amino acids, with masking applied to both the structure and sequence simultaneously. The same residues are used for both prediction and masking, with $\mathcal{T}_s = \mathcal{M}_x$. Following the seq2seq approach, 10 to 40 of the amino acids are masked, using a mix of continuous spans and random positions. With equal probability, we either (i) mask the corresponding coordinates $\mathcal{M}_x = \mathcal{M}_s$ or (ii) retain the full structural information $\mathcal{M}_x = \emptyset$ to use it as guidance.

- **struct2seq.** Only the structural information from the structural dataset $\mathcal{D}_{\mathrm{struct}}$ is used to predict amino acids $\mathbf{s}_i$ at specific target positions $\mathcal{T}_s$. The input sequence data is completely disregarded, replaced by a series of unknown tokens $*$, leaving only the structural information

x. The target positions for amino acid prediction and masked structures, $\mathcal{T}_s = \mathcal{M}_x$, are chosen using the same distribution as in the seq2seq task, alternating between continuous spans and random positions.

## 3 EMPIRICAL RESULTS

In this section, we evaluate the impact of incorporating structural information into the pre-training of antibody LLMs. Our evaluation focuses on three tasks: (i) sequence recovery of the variable region, (ii) editing of the CDR regions, and (iii) inverse folding. We compare the performance of IgBlend with five existing open-source antibody and nanobody language models, including AbLang (Olsen et al., 2022b), AbLang2 (Olsen et al., 2024), AntiBERTy (Ruffolo et al., 2021), Sapiens (Prihoda et al., 2022) and Nanobert (Hadsund et al., 2024) as well as two inverse folding models, including AntiFold (Høie et al., 2023) and ESM-IF (Hsu et al., 2022).

### 3.1 SEQUENCE RECOVERY

First, we evaluated the task of recovering missing residues in the variable region of an antibody. This task is particularly relevant for various applications where the goal is either to recover, edit, or mutate specific amino acids within a sequence. Following the benchmark established in (Olsen et al., 2022b; 2024), we proceeded as follows. First, we sampled 1,000 sequences/structures pairs, $(\mathbf{s}, \mathbf{x})$, per chain type from the test distribution, using Equation (1). Then, for each pair $(\mathbf{s}, \mathbf{x})$, we randomly sample a sequential mask $\mathcal{M}_s$ that contains between 10% and 40% of the residue indices from the full sequence $[1, \ldots, |\mathbf{s}|]$. To evaluate the benefits of incorporating structural information, we compared the performance of IgBlend using different input types. Specifically, we assessed each model's ability to recover the full sequence under various conditions: for sequence-only models, we used $\widehat{s} = \mathrm{Model}(\mathbf{s}_{/\mathcal{M}_s})$; for structure-guided sequential models, we used $\widehat{s} = \mathrm{Model}(\mathbf{s}_{/\mathcal{M}_s}, \mathbf{x})$; for sequential models that use masked structural information, we used $\widehat{s} = \mathrm{Model}(\mathbf{s}_{/\mathcal{M}_s}, \mathbf{x}_{/\mathcal{M}_s})$; and for inverse folding models, we used $\widehat{s} = \mathrm{Model}(\mathbf{x})$. For each chain type and each region reg∈[FW1, CDR1, FW2, CDR2, FW3, CDR3, FW4], where FW and CDR refer to framework and CDR regions of the chain respectively, we recorded the empirical accuracy of the models by computing the Levenshtein($\{\mathbf{s}_i, i \in \mathcal{M}_s \cap \mathrm{reg}\}, \{\widehat{\mathbf{s}}_i, i \in \mathcal{M}_s \cap \mathrm{reg}\}$) distance over each region and averaged the results over 1,000 sequences. Results are reported in Table 2 for the CDR3 regions and the remaining regions can be found in Table 4 of the Appendix. A few remarks are of order:

- First, sequence-only models (AbLang, AbLang2, AntiBERTy, Sapiens, IgBlend) performed similarly across all regions, with at most a 3% difference in accuracy between the models in most regions. In this sense, it has to be noted that IgBlend, trained using a combination of three different objectives shown in Equation (2), performed on par with models trained solely on the seq2seq task, indicating that multi-modal training does not compromise performance on individual tasks.

- Secondly, it is important to note that the performance of IgBlend consistently improves with the addition of more input modalities across all chain types. Specifically, for each chain type, IgBlend shows the same ranking in recovery rate: IgBlend(seq+struct guided) > IgBlend(seq+masked struct) > IgBlend(seq-only) and IgBlend(seq+struct guided) > IgBlend(inverse folding). Hence, adding information helps the model to be more precise and we deduce that proposed training procedure allows us to merge both modalities successfully.

- Most notably, by incorporating structural information alongside the masked sequence (IgBlend(seq+struct guidance)), we achieved consistently better results than sequence-only models across all regions and for all modalities. This improvement was particularly notable in the CDR3 region of the nanobody (N-CDR3), where IgBlend(seq+struct) outperformed the second-best model by 11.8% in accuracy. Similar improvements were observed in the CDR3 regions of the light chain (L-CDR3) with a 6.6% increase, and the heavy chain (H-CDR3) with a 7.7% increase.

| Mode | Model | Heavy | | | Light | | | Nanobody | | |
|---|---|---|---|---|---|---|---|---|---|---|
| | | CDR1 | CDR2 | CDR3 | CDR1 | CDR2 | CDR3 | CDR1 | CDR2 | CDR3 |
| **Sequence Only** | AbLang | **84.12** | 80.44 | 53.13 | 74.60 | 72.68 | 66.62 | 44.83 | 44.84 | 21.69 |
| | AbLang2 | 83.79 | **80.50** | **53.82** | **75.40** | 72.01 | 68.06 | 44.52 | 43.83 | 20.71 |
| | Antiberty | 83.72 | 80.30 | 48.37 | 75.12 | 72.75 | **68.21** | 46.16 | 47.29 | 25.63 |
| | Sapiens | 81.65 | 76.90 | 48.76 | 72.41 | 69.45 | 63.29 | 45.87 | 42.66 | 19.41 |
| | Nanobert | 56.22 | 42.58 | 25.31 | 7.76 | 05.64 | 06.98 | **64.20** | 61.43 | 33.09 |
| | IgBlend | 83.80 | 80.07 | 51.91 | 74.63 | **73.79** | 67.37 | 63.83 | **62.68** | **37.37** |
| **Inverse Folding** | Antifold | 76.73 | 71.53 | 36.27 | 59.04 | 59.85 | 46.79 | 45.48 | 44.40 | 23.50 |
| | ESM-IF | 50.08 | 46.74 | 20.27 | 34.59 | 45.00 | 31.59 | 31.01 | 41.56 | 16.10 |
| | IgBlend | **88.15** | **84.88** | **53.35** | **78.26** | **82.42** | **73.01** | **71.49** | **71.33** | **44.42** |
| **Seq + Masked Struct** | IgBlend | 85.00 | 80.76 | 54.07 | 75.46 | 75.61 | 69.11 | 67.77 | 64.23 | 40.05 |
| **Seq + Struct Guided** | IgBlend | **88.98** | **85.50** | **61.50** | **79.16** | **83.70** | **74.66** | **72.90** | **73.43** | **49.50** |

Table 2: **Sequence recovery results.** The task involves masking a proportion of residues in a sequence and having the model predict them. The table shows the average percentage of successfully recovered masked residues by region and chain type, with bold font highlighting the best results in each category. IgBlend in "Seq + Struct Guided" mode demonstrates the best overall performance.

## 3.2 COMPLEMENTARITY-DETERMINING REGION (CDR) EDITING

Second, we focused on the task of editing/recovering the CDR regions of a single chain, which is of particular importance in the process of optimizing antibodies for affinity. In this task, one of the CDR regions is randomly selected and fully masked, i.e., $\mathcal{M}_s \in \{\text{CDR1}, \text{CDR2}, \text{CDR3}\}$, and the models are asked to predict the residues within the selected fully masked CDR. Using the same experimental setup with the masked CDR $\mathcal{M}_s$, we evaluated the models—seq-only, structure-guided, and inverse folding—on 1,000 sequences sampled from the test set as described in Equation (1). These sequences were not seen during the training of IgBlend, and we recorded the percentage of successfully recovered residues for each chain type. The results can be found in the table shown in Figure 3 with all models being evaluated on the same masked sequences. To further evaluate how well models using structural information adhere to structural instructions, we assessed the structural similarities between the generated sequences and the input structure $\mathbf{x}$. We compared the top models in each category: AbLang2 for heavy/light chains, NanoBert for nanobodies, and AntiFold for inverse folding. We sampled 500 sequences per chain type from the test distribution and tasked each model with recovering a missing CDR. For each recovered sequence $\hat{s}$, we computed its structural approximation $\hat{\mathbf{x}} = \text{IgFold}(\hat{s})$ using IgFold with PyRosetta refinement. We then measured the Levenshtein($\{\mathbf{s}_i, i \in \mathcal{M}_s\}, \{\hat{\mathbf{s}}_i, i \in \mathcal{M}_s\}$) distance in the masked region and the RMSD($\{\mathbf{x}_i, i \in \mathcal{M}_s\}, \{\hat{\mathbf{x}}_i, i \in \mathcal{M}_s\}$) between the original and recovered structures to assess structural similarity. Results are shown in Figure 3, with extended findings available in Appendix E.2. From both evaluations, key observations include:

- First, as in previous experiments, the top-performing sequence-only models (AbLang, AbLang2, AntiBERTy, IgBlend) showed similar performance across different CDR regions. However, IgBlend outperformed the best sequence-only model by over 9% in accuracy for nanobodies. Notably, incorporating additional information consistently improved IgBlend's performance across all chain types (i.e., IgBlend (Seq+Struct Guided) > IgBlend (Seq+Masked Struct) > IgBlend (Seq-only)). Specifically, adding structural information to the masked sequence (IgBlend (seq+struct guidance)) significantly enhanced performance compared to the best sequence-only models, with improvements of 11.8% in H-CDR3, 6.74% in L-CDR3, and 15.43% in N-CDR3.

- Second, unlike the previous experiments, IgBlend (seq+struct guidance), which uses both $(\mathbf{s}_{\mathcal{M}_s}, \mathbf{x}_{\mathcal{M}_x})$, shows performance closer to IgBlend (inverse folding), which relies solely on the structure $\mathbf{x}$, than to IgBlend (seq-only), which depends only on the sequential information $\mathbf{s}_{\mathcal{M}_s}$. This suggests that, in the task of re-editing complete CDR regions, IgBlend relies more on structural information than on sequential data.

- In terms of structural similarity, it is noteworthy that IgBlend (structure guided) consistently achieves the highest percentage of structures within the bins corresponding to the smallest RMSD values for each chain type, surpassing even the best inverse folding model. Specifically, IgBlend shows 37%, 46%, and 51% of structures in the smallest RMSD bins (1 Å for heavy chains, 0.2 Å for light chains, and 1 Å for nanobodies), compared to 32%, 5%, and 26% for AntiFold in the same bins. See Figure 3 for a detailed comparison. Thus, in addition to outperforming sequential models, IgBlend(seq+struct guided) excels at generat-

| Mode | Model | Heavy | | | Light | | | Nanobody | | |
|---|---|---|---|---|---|---|---|---|---|---|
| | | CDR1 | CDR2 | CDR3 | CDR1 | CDR2 | CDR3 | CDR1 | CDR2 | CDR3 |
| **Sequence Only** | **AbLang** | 82.97 | **80.53** | 41.68 | 72.21 | 69.27 | 67.47 | 43.73 | 45.09 | 20.90 |
| | **AbLang2** | 82.85 | 80.31 | 41.62 | 72.94 | 69.66 | 68.03 | 43.05 | 41.43 | 20.16 |
| | **Antiberty** | 82.90 | 80.37 | 41.23 | 72.64 | 69.20 | 68.61 | 40.48 | 47.76 | 23.12 |
| | **Sapiens** | 81.44 | 77.13 | 38.45 | 71.18 | 67.22 | 63.03 | 44.25 | 39.99 | 19.79 |
| | **Nanobert** | 57.33 | 40.00 | 24.02 | 10.16 | 08.53 | 07.22 | 60.49 | 61.09 | 29.08 |
| | **IgBlend** | **83.15** | 80.33 | **41.84** | **73.14** | **69.79** | **68.70** | **62.58** | **63.81** | **29.53** |
| **Inverse Folding** | **AntiFold** | 75.41 | 70.99 | 36.97 | 57.05 | 58.98 | 49.12 | 44.70 | 44.92 | 22.02 |
| | **ESM-IF** | 49.90 | 44.19 | 19.65 | 33.68 | 43.70 | 31.46 | 30.74 | 39.98 | 15.34 |
| | **IgBlend** | **86.18** | **84.44** | **52.69** | **76.69** | **82.03** | **73.9** | **69.72** | **72.58** | **43.77** |
| **Seq + Masked Struct** | **IgBlend** | 84.00 | 80.61 | 43.37 | 74.00 | 73.10 | 70.61 | 65.93 | 64.75 | 32.28 |
| **Seq + Struct Guided** | **IgBlend** | **87.27** | **85.04** | **53.65** | **77.08** | **83.59** | **75.44** | **71.40** | **73.52** | **44.96** |

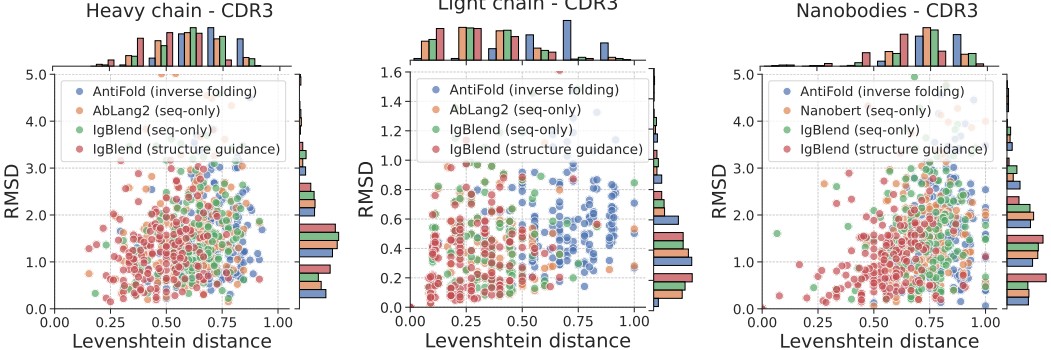

Figure 3: **CDR in-filling results:** One CDR region (CDR1, CDR2, or CDR3) is fully masked, and the model attempts to recover it. **Top:** The table shows the average percentage of correctly recovered residues for heavy chain (H), light chain (L) and nanobodies (N). **Bottom:** The graphs show Levenshtein distances of generated CDR3 regions from the original sequence on the x-axis, and RMSD of the predicted structures from the original backbone on the y-axis for each chain type.

ing sequences that can more accurately fold to the original backbone structure compared to those produced by AntiFold.

## 3.3 INVERSE FOLDING

Finally, we assessed IgBlend's ability to perform the inverse folding task (Hsu et al., 2022; Høie et al., 2023), which involves recovering a sequence **s** from its structure **x** alone. As with previous experiments, we sampled 500 structure-sequence pairs $(\mathbf{s}, \mathbf{x})$ for each chain type from the test set, which was not seen during IgBlend's training. Each inverse folding model was then asked to predict the sequence $\widehat{\mathbf{s}} = \text{Model}(\mathbf{x})$ based solely on the structure **x**. To evaluate model performance, we tested different temperatures: $T = 1e - 4$ for the highest probability sequence, $T = 1$ for unbiased results, and $T = 2$ and $T = 3$ for more diverse sequences. We recorded the normalized Levenshtein$(\mathbf{s}, \widehat{\mathbf{s}})$ distance between the predicted and original sequences, and the RMSD$(\mathbf{x}, \widehat{\mathbf{x}})$ of the approximated structure $\widehat{\mathbf{x}} = \text{IgFold}(\widehat{\mathbf{s}})$ as a measure of structural similarity. Results for the lowest temperature are shown in Figure 4, with additional details in Table 5 and Figure 7 in the Appendix. Key observations include:

- First, we observe a positive correlation between Levenshtein distance and RMSD for every model: as the sequence diverges more from the original (larger Levenshtein distance), the RMSD tends to increase, indicating a trade-off between sequence diversity and structural precision. Consequently, as temperature is increased to generate more diverse sequences, the RMSD increases. However, IgBlend demonstrates greater robustness to temperature changes. Second, all models perform better on light chains compared to heavy chains and nanobodies, suggesting that the inverse folding task is more challenging for heavy chains and nanobodies.

- Second, across all temperatures, chain types, and RMSD thresholds (see Figure 4 and Table 5 in the Appendix), the models consistently rank as follows based on the number of

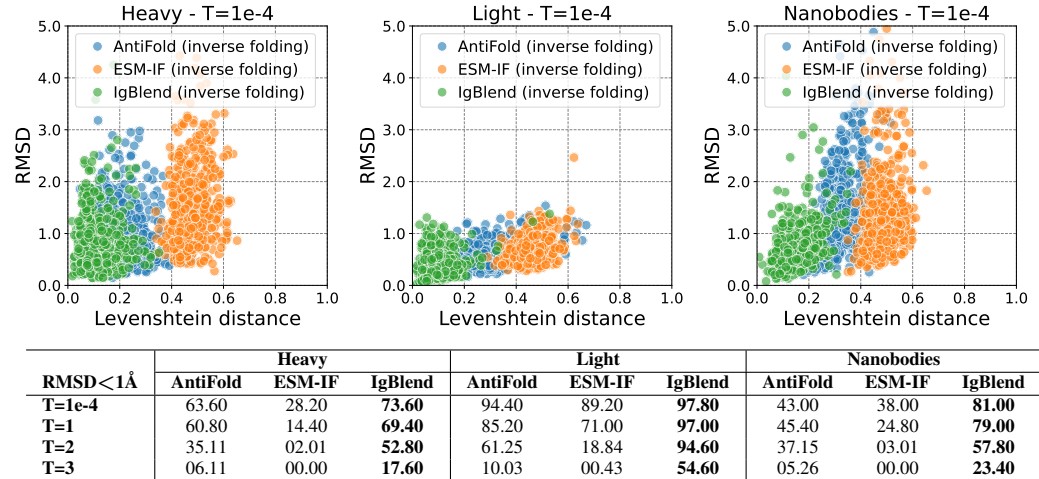

| RMSD<1Å | Heavy | | | Light | | | Nanobodies | | |
|---|---|---|---|---|---|---|---|---|---|
| | AntiFold | ESM-IF | IgBlend | AntiFold | ESM-IF | IgBlend | AntiFold | ESM-IF | IgBlend |
| T=1e-4 | 63.60 | 28.20 | **73.60** | 94.40 | 89.20 | **97.80** | 43.00 | 38.00 | **81.00** |
| T=1 | 60.80 | 14.40 | **69.40** | 85.20 | 71.00 | **97.00** | 45.40 | 24.80 | **79.00** |
| T=2 | 35.11 | 02.01 | **52.80** | 61.25 | 18.84 | **94.60** | 37.15 | 03.01 | **57.80** |
| T=3 | 06.11 | 00.00 | **17.60** | 10.03 | 00.43 | **54.60** | 05.26 | 00.00 | **23.40** |

Figure 4: **Inverse Folding Results:** In this task, the sequence is fully masked, and the model attempts to recover it from the structure. The top graph shows the normalized Levenshtein distance between generated and original sequences, with the y-axis displaying the RMSD of the generated structures relative to the original; greater spread in Levenshtein distance and lower RMSD indicate better performance. The bottom table lists the percentage of times each method produced a sequence with RMSD < 1 Å across 500 samples per modality. More details are available in Table 5 in the Appendix.

samples with RMSD below the threshold: IgBlend> AntiFold > ESM-IF. This shows that IgBlend outperforms current state-of-the-art methods in generating sequences that accurately fold back to the original structure. Notably, IgBlend is the first inverse folding model to achieve results on nanobodies comparable to heavy chains. However, this high accuracy comes with lower Levenshtein distance, a limitation seen in all tested settings.

### 3.4 HUMAN EPIDERMAL GROWTH FACTOR RECEPTOR (HER2): H-CDR3 DESIGN

Finally, we assessed the ability of our models to score sequences for H-CDR3 design using experimental data. We followed the approach outlined by Shanehsazzadeh et al. (2023), where deep learning models were used to re-design H-CDR3 sequences targeting HER2. In particular, they selected the therapeutic antibody trastuzumab, which targets HER2, as a template and re-designed the heavy chain CDR3 sequences $s$, conditioned on the HER2 antigen backbone structure from PDB 1N8Z and the trastuzumab framework sequences. The $K_D$ values of the generated sequences were then measured using a Fluorescence-activated Cell Sorting (FACS)-based ACE assay. To evaluate how well our models could pre-screen sequences likely to exhibit strong binding, we scored the sequences they generated (keeping the same H-CDR3 length as trastuzumab) by computing their log-likelihood using both AbLang2 and IgBlend (seq-only). To assess whether structural information improves scoring, we also scored the sequences using IgBlend (inverse folding), where the backbone structure $x$ of trastuzumab was provided as input, steering the model to favor sequences with a structure similar to trastuzumab and AntiFold. The $-\log(K_D)$ values against the likelihood scores from each model are displayed in Figure 5 as well as the Spearman correlation and Kendall tau.

- First, it is evident that sequence-only models, such as AbLang2 and IgBlend (seq-only), offer limited utility in predicting which sequences are likely to have strong binding affinities. The near-zero correlations indicate little to no relationship between the scores and the actual binding affinities. This can be attributed to the fact that these models are trained on sequence data alone, with no information about conformational states that would favor binding to HER2. As a result, they tend to generate biologically plausible sequences without prioritizing binding affinity (since much of the training set is derived from the OAS dataset, which lacks binding information).

- In contrast, structure-guided models such as IgBlend (inverse folding) and AntiFold demonstrate a stronger positive correlation between binding affinity and scoring, indicating their

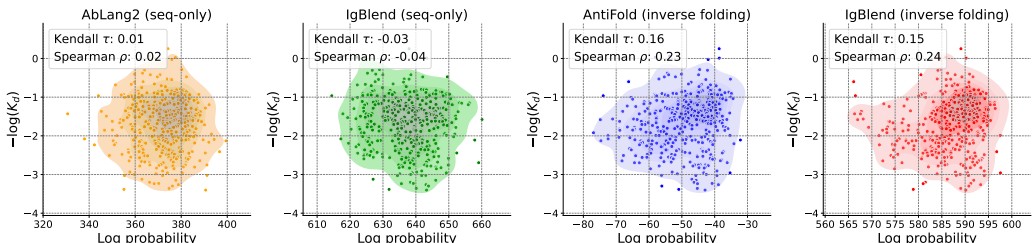

Figure 5: **HER2 H-CDR3 editing.** Each model scores sequences using their log-likelihood. The scatter plot displays the log-probability (score) of each sequence on the x-axis and the $-\log(K_D)$ values on the y-axis. Additionally, the density of the point cloud is displayed.

greater accuracy in identifying high-affinity sequences. IgBlend (inverse folding), which relies solely on backbone structure, further emphasizes the critical role of structural context in guiding models toward more favorable binding configurations.

This contrast highlights that while sequence-only models are effective at generating biologically viable sequences, structure-based models are superior for evaluating binding affinity, emphasizing the need to integrate structural data for more accurate predictions in H-CDR3 design.

## 4 CONCLUSION AND FUTURE WORK

In this study, we investigated how incorporating structural information into antibody LLMs enhances performance. We introduced a model that integrates both structural and sequential data, showing that this combination consistently improves performance across all benchmarks compared to sequence-based and inverse-folding models. However, while effective, our approach sometimes sacrifices sequence diversity for accuracy. Future work will focus on including side-chain information and expanding structural datasets.

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
