## A BACKGROUND ON ANTIBODIES

**Background.** In humans, antibodies are classified into five isotypes: IgA, IgD, IgE, IgG, and IgM. This work primarily focuses on IgG antibodies, which are Y-shaped glycoproteins produced by B-cells (see Figure 1), as well as nanobodies, which are antibody fragments consisting of a single monomeric variable domain. Henceforth, "antibody" will specifically refer to IgG antibodies. Antibodies consist of distinct regions that play specific roles in the immune response. The Fab (fragment antigen-binding) region, composed of both variable (V) and constant (C) domains from the heavy and light chains, is primarily responsible for antigen binding. Within this region, the antigen-binding site is formed by the variable domains — VH for the heavy chain and VL for the light chain — which determine the specificity of the antibody and enable it to recognize and bind to specific antigens. The Fv (fragment variable) region is the smallest functional unit of an antibody that can still bind to an antigen. It consists solely of the variable domains (VH and VL) of the heavy and light chains, without the constant domains. Within the variable domains, there are two key distinct regions: the framework regions and the complementarity-determining regions (CDRs). The framework regions provide structural support, maintaining the overall shape of the variable domains, while the CDRs, comprising three loops on both the VH and VL chains, are directly involved in binding to the antigen. These CDRs are crucial for the precise recognition and interaction with specific antigens. While the Fv region is essential for the initial recognition and binding of antigens, it lacks the effector functions present in the full antibody. The Fab region, being larger and more complex due to the inclusion of both variable and constant domains, is generally more stable and has a higher affinity for antigens. The Fv region, on the other hand, is simpler and more easily engineered for various applications, such as in the development of single-chain variable fragment (scFv) antibodies. The base of the Y-shaped antibody, known as the Fc (fragment crystallizable) region, is involved in regulating immune responses. It interacts with proteins and cell receptors, ensuring that the antibody generates an appropriate immune response. Moreover, nanobodies, which are small, single-domain antibodies derived from heavy-chain-only antibodies found in certain animals such as camels and llamas, are even more compact than traditional Fv regions. They retain full antigen-binding capacity while offering advantages such as increased stability and easier production, making them valuable tools in both therapeutic and diagnostic applications.

## B ARCHITECTURAL DETAILS

In Table 3, we collect the full architectural details of the IgBlendarchitecture used in the paper.

## C DATA SET

**Data source.** To create a model capable of processing both sequential and structural information, we needed to address the significant asymmetry in the availability of data across these modalities (204M sequences and 3M structures as shown in Table 1). Therefore, we compiled two datasets: (1) a structural dataset $\mathcal{D}_{\text{struct}}$, which includes structures paired with their corresponding sequences, and (2) a sequential dataset $\mathcal{D}_{\text{seq}}$, which consists solely of sequence data. These datasets were derived from four primary sources: SAbDab (Dunbar et al., 2014), which contains experimentally determined structures using techniques such as electron crystallography and X-ray diffraction; PLAbDab (Abanades et al., 2023a), which provides sequences derived from patents; OAS datasets (Olsen et al., 2022a), which compile and annotate immune repertoires; and INDI (Deszyński et al., 2022), which contains sequences of nanobodies. Given the relatively small number of experimentally determined structures (e.g., approximately 2,000 samples from SAbDab, as shown in Table 1 after applying our selection criteria), we expanded our structural dataset by incorporating inferred structures. In addition to the inferred structures already present in the PLAbDab dataset (folded with ImmuneBuilder), we generated additional structures from the OAS paired, unpaired and INDI. The paired sequences from OAS were folded with ImmuneBuilder (Abanades et al., 2023b) and a clustered version of the unpaired OAS and INDI dataset were folded using IgFold (Ruffolo et al., 2023). This process resulted in approximately 4 million unique structures. For the sequential dataset, we extracted data from four repertoires: OAS paired, OAS unpaired, PLAbDab paired, PLAbDab unpaired and INDI.

For each of the datasets $\mathcal{D}_{\text{struct}}$ and $\mathcal{D}_{\text{seq}}$, we begin by removing all duplicates, defined as pairs of data with identical sequences. Next, only the data that meet the following criteria are retained:

| Structure module | value |
|---|---|
| gvp_eps | 0.0001 |
| gvp_node_hidden_dim_scalar | 512 |
| gvp_node_hidden_dim_vector | 256 |
| gvp_num_encoder_layers | 4 |
| gvp_dropout | 0.1 |
| gvp_encoder_embed_dim | 512 |
| transformer_encoder_layers | 2 |
| encoder_embed_dim | 512 |
| transformer_dropout | 0.1 |
| encoder_attention_heads | 8 |
| encoder_ffn_embed_dim | 1024 |
| Sequence Module | |
| d_model | 512 |
| dropout | 0.1 |
| layer_norm_eps | 0.0001 |
| nhead | 8 |
| activation | SwiGLU |
| dim_feedforward | 512 |
| layer_norm_eps | 0.0001 |
| Multi-modal encoder | |
| d_model | 1024 |
| num_layers | 4 |
| n_head | 16 |
| dim_feedforward | 1024 |
| activation | SwiGLU |
| prediction_head | |
| d_model | 1024 |
| activation | GELU |

Table 3: Hyper-parameters of the IgBlendmodel.

(1) no unknown residues, (2) no missing residues, and (3) no shorter than expected IMGT regions (Ehrenmann et al., 2010), as determined by running ANARCI (Dunbar & Deane, 2016). After these cleaning steps, we are left with two datasets: $\mathcal{D}_{\text{struct}} = \{(\mathbf{s}, \mathbf{x})_1, \ldots, (\mathbf{s}, \mathbf{x})_{|\mathcal{D}_{\text{struct}}|}\}$, which contains pairs of sequences and structures, and $\mathcal{D}_{\text{seq}} = \{(s, *)_1, \ldots, (s, *)_{|\mathcal{D}_{\text{seq}}|}\}$, which contains only sequential information.

## D    TRAINING DETAILS

The model was trained on 8 A10G GPUs using a distributed DDP strategy and the PyTorch Zero Redundancy Optimizer (Rajbhandari et al., 2020). The total number of training steps was predetermined at 125,000. The learning rate was warmed up over the first 200 steps to a peak of 0.001, after which it was gradually reduced to zero using a cosine scheduler. Training was conducted in 16-bit precision. To conserve memory and enable a larger batch size, gradient activation checkpointing was implemented immediately after the structural module. The effective batch size was set to 90 per GPU, resulting in a total batch size of 720 samples per step. The AdamW optimizer was used with a weight decay parameter of 0.1, epsilon of 0.00001, and betas of [0.9, 0.95] for regularization. More details about the hyperparameters can be found in the Appendix B.

## E    EXPERIMENTAL RESULTS

### E.1    SEQUENCE RECOVERY

Table 4 records the sequence recovery rate on all regions and for each modality.

| Heavy chains | FW-1 | CDR-1 | FW-2 | CDR-2 | FW-3 | CDR-3 | FW-4 |
|---|---|---|---|---|---|---|---|
| AbLang (seq-only) | 95.65 | 84.12 | 93.49 | 80.44 | 92.22 | 53.13 | 96.32 |
| AbLang2 (seq-only) | 95.54 | 83.79 | 93.67 | 80.50 | 92.21 | 53.82 | 96.16 |
| Antiberty (seq-only) | 95.71 | 83.72 | 93.24 | 80.30 | 92.15 | 48.37 | 96.27 |
| Sapiens (seq-only) | 94.23 | 81.65 | 91.13 | 76.90 | 89.21 | 48.76 | 95.31 |
| Nanobert (seq-only) | 74.48 | 56.22 | 72.97 | 42.58 | 65.39 | 25.31 | 85.17 |
| IgBlend(seq-only) | 95.66 | 83.80 | 93.25 | 80.07 | 91.91 | 51.91 | 96.23 |
| IgBlend(seq+masked struct) | 95.86 | 85.00 | 93.32 | 80.76 | 91.96 | 54.07 | 96.10 |
| IgBlend(seq+struct guided) | **96.52** | **88.98** | **95.38** | **85.50** | **93.68** | **61.50** | **97.15** |
| IgBlend(inverse folding) | 96.02 | 88.15 | 94.94 | 84.88 | 93.36 | 53.35 | 96.64 |
| Antifold (inverse folding) | 87.07 | 76.73 | 88.90 | 71.53 | 88.66 | 36.27 | 91.70 |
| ESM-IF (inverse folding) | 55.69 | 50.08 | 63.43 | 46.74 | 59.41 | 20.27 | 57.96 |
| Light chains | FW-1 | CDR-1 | FW-2 | CDR-2 | FW-3 | CDR-3 | FW-4 |
| AbLang (seq-only) | 93.18 | 74.60 | 88.55 | 72.68 | 92.70 | 66.62 | 93.31 |
| AbLang2 (seq-only) | 94.06 | 75.40 | 88.79 | 72.01 | 93.01 | 68.06 | 93.54 |
| Antiberty (seq-only) | 94.05 | 75.12 | 88.63 | 72.75 | 93.01 | 68.21 | 93.63 |
| Sapiens (seq-only) | 92.94 | 72.41 | 87.25 | 69.45 | 91.58 | 63.29 | 88.45 |
| Nanobert (seq-only) | 16.15 | 7.76 | 19.27 | 05.64 | 21.12 | 06.98 | 41.97 |
| IgBlend(seq-only) | 93.97 | 74.63 | 88.43 | 73.79 | 92.86 | 67.37 | 92.32 |
| IgBlend(seq+masked struct) | 94.00 | 75.46 | 89.17 | 75.61 | 93.00 | 69.11 | 94.10 |
| IgBlend(seq+struct guided) | **95.07** | **79.16** | **91.78** | **83.70** | **94.43** | **74.66** | **96.46** |
| IgBlend(inverse folding) | 94.37 | 78.26 | 91.19 | 82.42 | 93.89 | 73.01 | 95.59 |
| Antifold (inverse folding) | 68.86 | 59.04 | 76.40 | 59.85 | 84.69 | 46.79 | 75.08 |
| ESM-IF (inverse folding) | 56.32 | 34.59 | 63.63 | 45.00 | 64.52 | 31.59 | 51.89 |
| Nanobodies | FW-1 | CDR-1 | FW-2 | CDR-2 | FW-3 | CDR-3 | FW-4 |
| AbLang (seq-only) | 87.46 | 44.83 | 60.88 | 44.84 | 78.49 | 21.69 | 87.29 |
| AbLang2 (seq-only) | 87.21 | 44.52 | 60.58 | 43.83 | 78.07 | 20.71 | 87.94 |
| Antiberty (seq-only) | 87.10 | 46.16 | 74.53 | 47.29 | 85.09 | 25.63 | 95.85 |
| Sapiens (seq-only) | 88.65 | 45.87 | 60.35 | 42.66 | 75.58 | 19.41 | 86.01 |
| Nanobert (seq-only) | 93.44 | 64.20 | 86.92 | 61.43 | 88.32 | 33.09 | 97.12 |
| IgBlend(seq-only) | 93.35 | 63.83 | 87.40 | 62.68 | 88.40 | 37.37 | 97.24 |
| IgBlend(seq+masked struct) | 94.79 | 67.77 | 88.07 | 64.23 | 88.73 | 40.05 | 97.35 |
| IgBlend(seq+struct guided) | **96.45** | **72.90** | **92.26** | **73.43** | **91.94** | **49.50** | **97.72** |
| IgBlend(inverse folding) | 96.04 | 71.49 | 91.93 | 71.33 | 91.65 | 44.42 | 97.22 |
| Antifold (inverse folding) | 87.38 | 45.48 | 64.56 | 44.40 | 80.09 | 23.50 | 87.32 |
| ESM-IF (inverse folding) | 56.83 | 31.01 | 57.67 | 41.56 | 62.43 | 16.10 | 55.13 |

Table 4: **Sequence recovery results.** The task consists of masking randomly a proportion of residues within a sequence and asking the model to predict the masked residues. The table display the average percentage of successfully recovered masked residues in each region and for each type of chain. Bold font indicates the best result in the comparison

## E.2 CDR EDITING

Figure 6 collects the result of the CDR recovery experiment in all CDR regions.

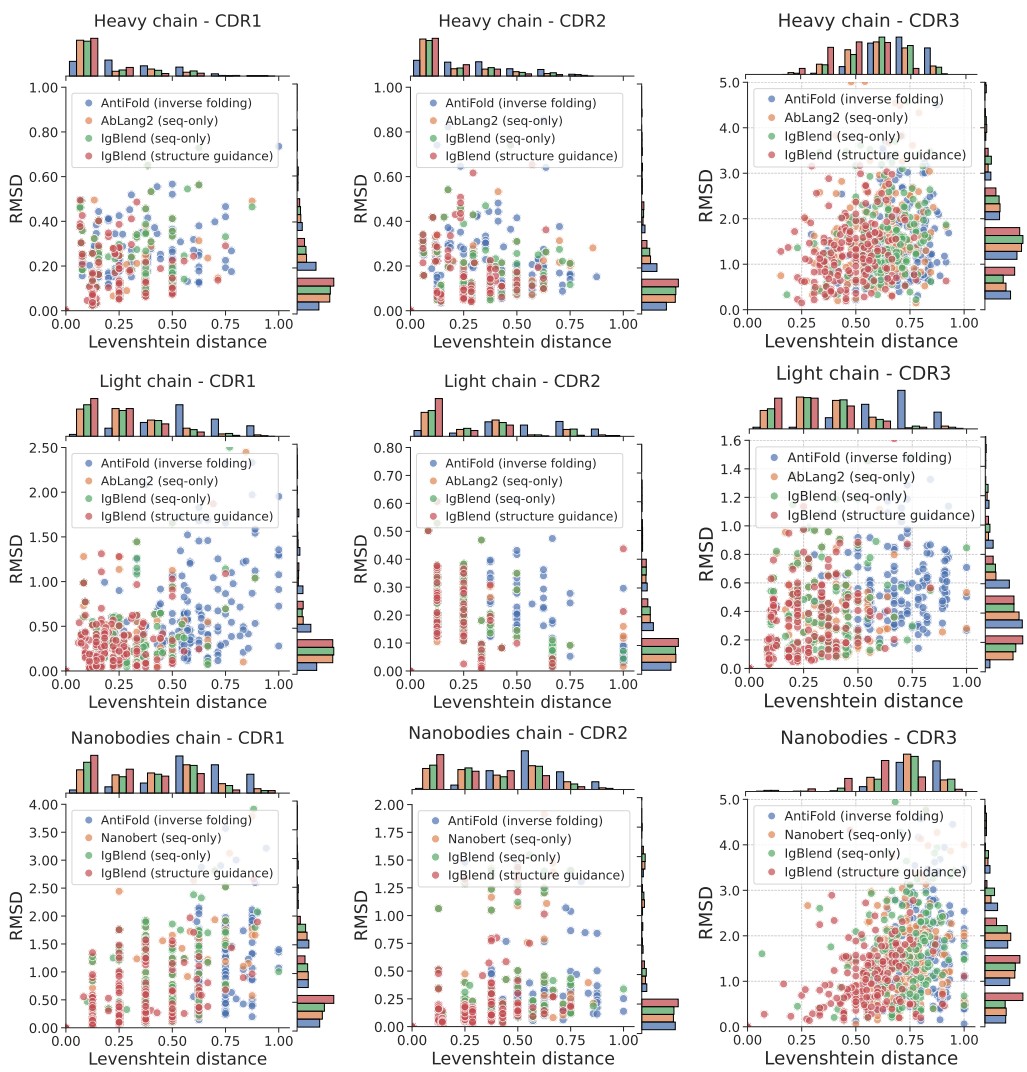

Figure 6: **CDR recovery results:** One series of aminco acid of the sequence is fully masked (one CDR), and the model attemps to recover it. AntiFold only uses the structural information. IgBlend (structure guidance) uses the masked sequence and the structure information. The distances (both Levenshtein and RSME) are only computed in the masked CDR regions. The x-axis displays the Levensthein distance of the generated sequences to the original one and the y-axis reports the RMSE of the generated sequence with regards to the original structure.

## E.3 INVERSE FOLDING

Figure 7 displays the inverse folding results for the different temperatures. Table 5 reports the results for different RMSD thresholds.

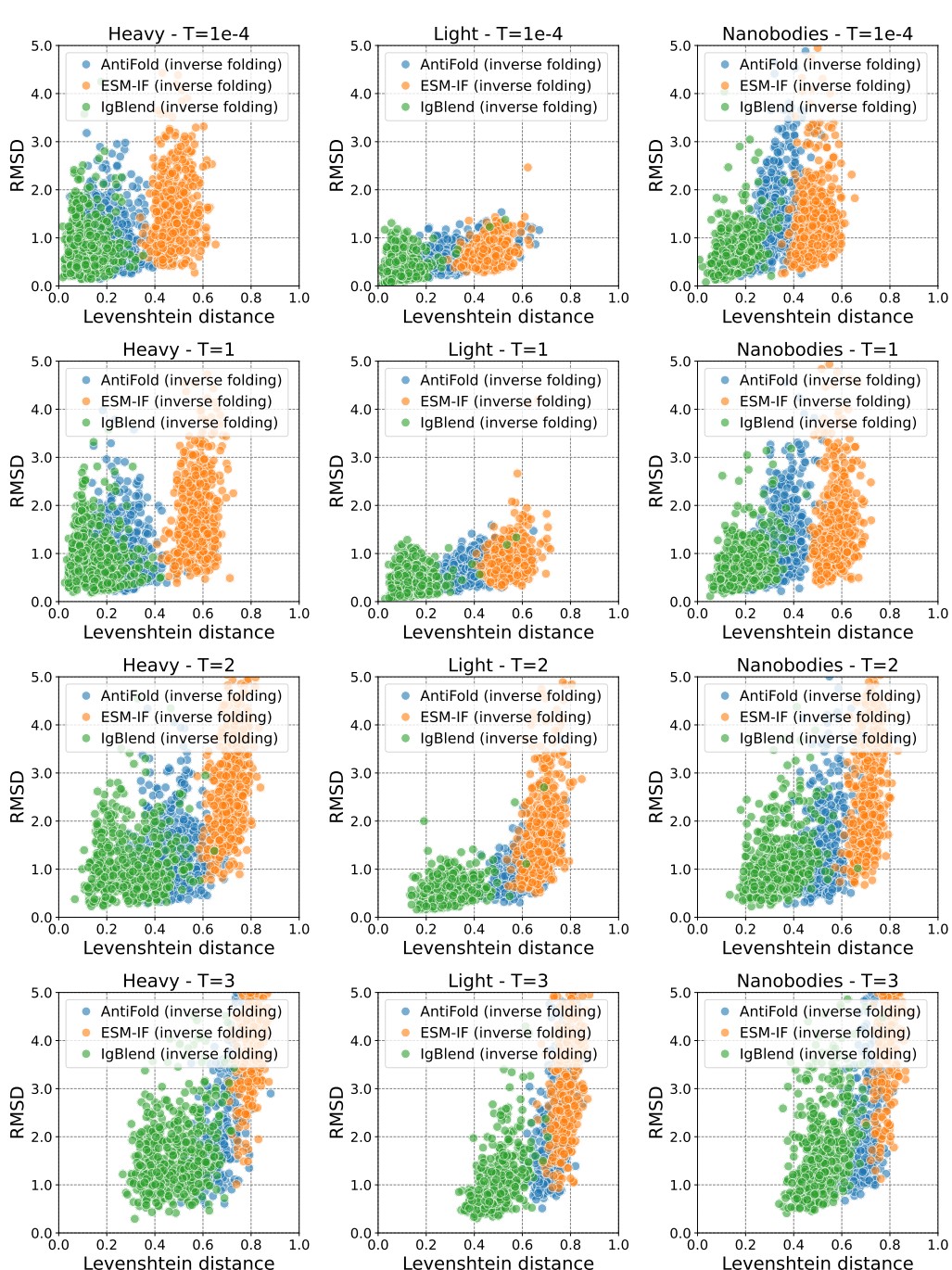

Figure 7: **Inverse folding results:** The sequence if fully masked, and the model attempts to recover it from the structure. **Top:** the graph displays the normalized Levenshtein distance of the generated sequences to the original sequences associated with the input structure and the y-axis reports the RMSD of the folded structure of the generated sequences with regards to the original structure set as input. For both metrics, lower is better.

| RMSD<0.5 | Heavy | | | Light | | | Nanobodies | | |
|---|---|---|---|---|---|---|---|---|---|
| | AntiFold | ESM-IF | IgBend | AntiFold | ESM-IF | IgBend | AntiFold | ESM-IF | IgBend |
| T=1e-4 | 20.40 | 02.40 | **27.6** | 49.20 | 24.20 | **72.0** | 07.20 | 03.40 | **32.00** |
| T=1 | 18.60 | 00.80 | **25.00** | 34.00 | 09.80 | **68.00** | 08.79 | 02.00 | **30.00** |
| T=2 | 05.95 | 00.00 | **10.60** | 07.50 | 00.00 | **47.40** | 04.75 | 00.00 | **08.80** |
| T=3 | 00.00 | 00.00 | **01.00** | 00.00 | 00.00 | **06.60** | 00.00 | 00.00 | **00.80** |
| **RMSD<1** | **AntiFold** | **ESM-IF** | **IgBend** | **AntiFold** | **ESM-IF** | **IgBend** | **AntiFold** | **ESM-IF** | **IgBend** |
| T=1e-4 | 63.60 | 28.20 | **73.60** | 94.40 | 89.20 | **97.80** | 43.00 | 38.00 | **81.00** |
| T=1 | 60.80 | 14.40 | **69.40** | 85.20 | 71.00 | **97.00** | 45.40 | 24.80 | **79.00** |
| T=2 | 35.11 | 02.01 | **52.80** | 61.25 | 18.84 | **94.60** | 37.15 | 03.01 | **57.80** |
| T=3 | 06.11 | 00.00 | **17.60** | 10.03 | 00.43 | **54.60** | 05.26 | 00.00 | **23.40** |
| **RMSD<1.5** | **AntiFold** | **ESM-IF** | **IgBend** | **AntiFold** | **ESM-IF** | **IgBend** | **AntiFold** | **ESM-IF** | **IgBend** |
| T=1e-4 | 84.60 | 57.80 | **89.60** | 99.80 | 99.80 | **100.0** | 62.60 | 67.60 | **95.60** |
| T=1 | 81.20 | 42.00 | **86.60** | 99.80 | 95.80 | **100.0** | 72.20 | 56.60 | **94.00** |
| T=2 | 66.17 | 11.85 | **78.40** | 92.50 | 46.69 | **99.40** | 62.85 | 17.27 | **80.60** |
| T=3 | 23.40 | 00.65 | **51.00** | 27.96 | 04.49 | **83.40** | 24.44 | 00.43 | **51.20** |
| **RMSD<2** | **AntiFold** | **ESM-IF** | **IgBend** | **AntiFold** | **ESM-IF** | **IgBend** | **AntiFold** | **ESM-IF** | **IgBend** |
| T=1e-4 | 94.00 | 77.80 | **96.40** | 100.0 | 99.80 | **100.0** | 78.20 | 84.80 | **98.20** |
| T=1 | 92.40 | 66.00 | **95.40** | 100.0 | 99.00 | **100.0** | 84.20 | 75.80 | **97.80** |
| T=2 | 84.68 | 27.91 | **90.80** | 98.12 | 68.34 | **99.60** | 77.97 | 34.14 | **90.00** |
| T=3 | 47.12 | 02.16 | **71.60** | 48.02 | 11.32 | **93.00** | 39.85 | 02.59 | **69.80** |

Table 5: **Inverse folding results:** The sequence if fully masked, and the model attempts to recover it from the structure. The table displays the percentage of sequences generated by each method with a RMSD below a given threshold and for different temperatures. Higher is better.