# OpenReview forum: "IgBleng: Unifying 3D structures and sequences in antibody language models"
_ICLR.cc/2025/Conference — Submitted to ICLR 2025_

### Official Review · Reviewer_r8CA · 2024-10-30

**Soundness:** 3
**Presentation:** 3
**Contribution:** 2
**Rating:** 3
**Confidence:** 4

**Summary:**

The paper introduces IgBlend, a large language model (LLM) specifically designed for antibody engineering, combining 3D structural data with sequential data. IgBlend aims to address the limitations of existing antibody LLMs by incorporating 3D backbone coordinates (C-alpha, N, and C atoms) alongside traditional sequence information. The model leverages a large dataset comprising over 4 million unique structures and 200 million unique sequences, enabling it to perform tasks like sequence recovery, CDR editing, and inverse folding. Empirical evaluations show that IgBlend consistently outperforms state-of-the-art antibody models across several benchmarks, and its log probabilities correlate well with binding affinity.

**Strengths:**

The proposed antibody language model has biomedical applications, specifically in antibody and therapeutic design, where such capabilities are highly impactful.

The model proposes an novel combination of structure and sequence information, which is theoretically a promising approach for antibody design tasks.

The methodological and architectural descriptions are mostly clear, and the integration of benchmark comparisons supports the claims effectively.

**Weaknesses:**

Lack of clarification of the benchmark dataset. Given the fact that the model has seen millions of training samples, I worry about the data leak in benchmarking and the author did not mention much about the benchmark construction.

The author compare the model on several CDR infilling tasks. As a language model, the utility in representation learning is unclear.

The HER2 H-CDR3 editing experiment shows a weak correlation (Spearman correlation: 0.24) between model scores and binding affinity, which is also close to the baseline AntiFold of 0.23.

Lack of open source code for the implementation and experimental results.

**Questions:**

What is the size of your language models?

Antibody sequences and structures are very humongous especially in the constant regions. I wonder if it's worth to train on millions of examples where most of them are similar. In addition, the introduction of predicted antibody structures in the training might introduce biases, and limits the model generalization capabilities. I would like to see some ablations the training data diversity, quantity, and the inclusion of predicted structures.

It would be convincing to also compare the model in representation tasks, such as antibody related downstream tasks.

It's important to show the performance gain is from the algorithm instead the memorization of the training set. To this end, I suggest authors to construct a non-redundant test set (such as sequence identity of 50%) for the infilling tasks.

---

> ### Author Response · Authors · 2024-11-19
> **Response to reviewer r8CA**
>
> First, we would like to thank the reviewer for its valuable feedback. We try to answer the main concerns below:
>
> - The language model has approximately 100M parameters
>
> - We do agree that it is an important point to get ablation studies on the training dataset such as data diversity, quantity and the inclusion of predicted structures. We would like to see such results in the literature as well. However, to the best of our knowledge, no paper discuss those aspects. It is probably to the cost of running such experiments.
>
> - Yes, the best experimental dataset we used was sequence ranking (Experiment X). In future works, we plan to fine tune several models on different downstream tasks to compare the model (humanization, developability and paratope prediction). However, due to the size of the paper, we plan to investigate all these problems in future works
>
> - We do agree that measuring the capacity of the algorithm rather than memorization is the end goal. Here we used the ratio used in other LLMs papers, which also present the same limitations (between memorizing and algorithm). We also thought about this issue while writing the paper. However, to have a fair comparison between all models, it is generally hard to have a dataset that is in the intersection of all models. But we do agree that using a 50% identity cut-off might be better.

---

> > ### Comment · Reviewer_r8CA · 2024-11-20
> > **Response**
> >
> > Thank you to the authors. I appreciate the clarification and I maintain the score.

---

### Official Review · Reviewer_LwS5 · 2024-11-02

**Soundness:** 2
**Presentation:** 2
**Contribution:** 2
**Rating:** 3
**Confidence:** 3

**Summary:**

This paper presents the IgBlend method, a pre-trained model for antibody modeling. The model can encode both sequence and structure information, and predict the masked sequence. The model can also be used in other tasks like inverse folding or CDR design.

**Strengths:**

-	The authors studied an important research topic.
-	The authors conducted experiments on many downstream tasks.

**Weaknesses:**

-	I have concerns about the novelty because jointly modeling sequence and structure have already been widely studied in many scientific tasks, such as organic small molecules [1,2] and protein [3].

-	In Table 2, for the sequence only settings, the IgBlend performance seems to be worse than baselines.

-	Minor: the caption of Fig.2 should below the figure.

[1] Dual-view Molecular Pre-training, KDD 2023

[2] One Transformer Can Understand Both 2D & 3D Molecular Data, ICLR 2023

[3] Protein sequence and structure co-design with equivariant translation, ICLR 2023

**Questions:**

-	Does authors have any plan to release the pretraining code, data, and checkpoints for reproducibility
-	As there are three coordinates for each position (C-alpha, N, and C), how are they processed by GVP-GNN? How are they combined?

---

> ### Author Response · Authors · 2024-11-19
> **Response to reviewer LwS5**
>
> First, we would like to thank the reviewer for its valuable feedback. We try to answer the main concerns below:
>
> - We do agree that jointly modeling different modalities have been shown to work in different domains (images/text) and even more close domains (small molecules/proteins). However, to the best of our knowledge, it is the first architecture work that shows how to design a full training strategy and models that actually work for antibodies and they display consistant improvements over the existing methods. Here, the main message from the paper is the following: 1) we can successfully create a multi-modal architecture for antibodies, 2) we provide an architecture that works for these modealities, 3) we show which data to use and 4) we provide the associated pre-training objectives
>
> - In Table 2, yes IgBlend does not always have the best results in sequence-only modalities. But, in our opinion, it is normal and it has to be noted that no model gets the best results every time, in the sense that there is no such things as the best model for sequence only model. However, in the main message of the paper, it is important to note that the approach we propose (sequence + structure) does provide the best results consistently over any sequence-only models
>
> - Thanks you for pointing out the position of the figure 2
>
> - We have plans to indeed release the weights of the model, and code
>
> - For the GVP, the three input coordinates of each residues are passed on a graph neural networks and the three are aggregated for each residue. The GVP considers the closest 20 residues around a given residue. It is a very technical architecture, and we strongly advise to read the details in [4] in case you are actively interested in all the technical details

---

### Official Review · Reviewer_cXNt · 2024-11-03

**Soundness:** 2
**Presentation:** 2
**Contribution:** 2
**Rating:** 3
**Confidence:** 4

**Summary:**

The authors present IgBlend, a model that combines antibody sequences and to enhance antibody engineering. The model shows improved performance in several benchmarks and shows strong correlation with binding affinities in experimental validation.

**Strengths:**

1. The authors presented an effective pre-training framework, unifying antibody sturctures and sequences.

**Weaknesses:**

1. The authors did not compare IgBlend with another similar method, LM-Design, despite mentioning it in the related work section.
2. The whole structure information ("Struct Guided") was used in sequence recovery and CDR editing tasks, which may introduce potential data leakage.
3. The sequence recovery and CDR editing tasks are quite similar, and the observed improvements appear marginal.
4. This paper is not clearly written, and there is a typo in the title (IgBleng -> IgBlend).

**Questions:**

Why did the authors choose GVP as the structure encoder instead of other models, such as MPNN?

---

> ### Author Response · Authors · 2024-11-19
> **Response to reviewer cXNt**
>
> First, we would like to thank the reviewer for its valuable feedback. We try to answer the main concerns below:
>
> - The focus of the experiments was on the antibodies so we mostly compared it with Antibody models (which are known to be better than protein only models)
>
> - To avoid data leakage, we used a clustering algorithm on both sequences and structure to create a train/test/val sets. We agree that this approach might have limitations, but to the best of our knowledge, it is used in most LLMs training papers and the best we can do right now. If you have any other suggestions, we can take them to solve this problem. One big issue notably comes from the fact that we have 4M structures and performing an exhaustive clustering on these structures is hard.
>
> - The observed improvement of order of magnitude of 10%. The main take-away message is that the improvement is constantly present with regards to the chain type: heavy, light and nanobodies. Thus, for application, it suggests that it is always better to capture both structure and sequence. Moreover, when using different-only LLMs, the performance difference are of magnitude 3%. Here, we show that using both sequence and structure (even synthetic) can be used in practical settings to improve significantly upon sequential only models, which is an important message in our opinion
>
> - IgBleng is indeed a typo. Thank you for pointing that out.
>
> - We used GVP instead of MPMM, simply because it was known to work for proteins and it indeed also works for Antibodies. Also, it has the property of being translation/rotation invariant as opposed (as far as we know) to MPMM. Lastly, keep also in mind that the runs are expensive, and we could not perform many tests

---

### Official Review · Reviewer_8yDb · 2024-11-03

**Soundness:** 2
**Presentation:** 2
**Contribution:** 2
**Rating:** 5
**Confidence:** 4

**Summary:**

This paper introduces IgBlend, a new method which adds structural information to an
antibody language model using a Geometric Vector Perceptron. The authors describe a
pre-training approach which combines multiple objectives for learning sequence and
structure-based tasks.
Currently, I would recommend to reject this paper for the following reasons: (1) The novelty
of the approach is limited. There have already been several models proposed which
integrate structure into protein language models, including LMDesign which is cited by the
authors, and the model is very similar to ESM-IF which has already been adapted to
antibodies (2) The comparison to other methods is not fair. In particular, other methods
were not designed to be run on single chains and/or Nanobodies. (3) The apparent
inclusion of modelled structures in the test set means that the results cannot be trusted
since IgBlend may simply be learning bias in IgFold, especially for Nanobodies.

**Strengths:**

• The paper is well-written and it is easy to understand the differences between the
experimental settings.
• The comparison to sequence-only settings throughout demonstrates that this
model operates effectively as a sequence model in the absence of structure.
• The analysis of model-guided HER2 design was interesting and motivated the
inclusion of structure.
• The ablation of the different model inputs is helpful and informative.

**Weaknesses:**

• The AntiFold results are much worse than those quoted in the AntiFold paper. This
is likely because AntiFold was trained using both chains whereas this paper tests it
on individual chains (including Nanobodies) which is not a fair comparison.
• The test set seems to include a large number of IgFold-generated structures. If this
is the only method trained on IgFold structures, then it is very likely to perform
better by learning bias in IgFold. Previous approaches test only on high-quality
experimental structures. This point holds especially when comparing RMSDs to
IgFold-predicted structures.
• “Notably, IgBlend is the first inverse folding model to achieve results on nanobodies
comparable to heavy chains” - this claim is not well supported since it appears that
all tested Nanobody structures were modelled.
• The authors note the similarity to LM-Design in the introduction but do not provide a
comparison in the experiments.
• The biggest advantage of this work compared to the extensive work on antibody
language modelling and inverse folding is the “Seq + Struct Guided” setting,
however the benefit of this is not demonstrated in any experiments except
pretraining. Could this be used to generate better embeddings or in a realistic
antibody design task?

**Questions:**

• I assume the title IgBleng is a typo?
• Is there some inherent benefit to language models which justifies the exclusion of
tools such as AbMPNN?

---

> ### Author Response · Authors · 2024-11-19
> **Response to reviewer 8yDb**
>
> First, we would like to thank the reviewer for its valuable feedback. We try to answer the main concerns below:
>
> - First , we would like to highlight the fact that it is generally hard to have a fair comparison between deep learning systems (see, e.g, LLMs in NLP) as they often exploit different training datas (bias), methodologies (type of inputs) and objectives (pre-training objectives). For instance, here, IgBlend is only trained to inverse-fold unique sequences while AntiFold is trained on paired data. So there is no middle ground to compare them “fairly”. In one case, e.g., single chain, AntiFold will be biased and in the other case, paired sequences, IgBlend will be biased. To get an OK comparison, we only tried here both methods in a test set not seen during training of IgBlend, which only contains single chains (that also supports nanobodies)
>
> - Second, it should be noted that IgBlend has been trained on both ImmuneBuilder structures (OAS paired) and IgFold structures (OAS unpaired and nanobodies). There might indeed be a bias in the comparison due to the use of different algorithms for folding. The only way we see to remove this bias would be to retrain the models on exactly to the same dataset, which is however impossible due to the cost of the experiments. However, in the set of experiments, we tried the middle ground by answering the following question: “Given a sequence (without any specific) structure, what is the best algorithm to get a sequence with similar to the structure” that can encompass every chain type: heavy, light and nanobodies. However, we do agree with the reviewer that the experiments are not “perfect”, which is almost impossible to get. One possible way would be to test on a them on experimentally determined structures, but it is hard to get the intersection of test sets of three different methods
>
> - We agree with the statement “Notably, IgBlend is the first inverse folding model to achieve results on nanobodies comparable to heavy chains” is over general. In the paper we now say that it is the best method for at least synthetic structures.
>
> - Actually, the focus of the experiments was on antibodies, so we mostly compared it with Antibody models, which are known to be much better than generic protein models on antibodies tasks.
>
> - In our opinion, the extra embeddings using the sequence+structure mode can be seen in the CDR3 experiments, where it shows that filled sequences using this mode tend to fill the CDR3 in a better way. For more experimental evaluation, we did not find a dataset where it could be used directly, but we strongly believe that it is of particular interest for applied settings. In future work, we will fine-tune the different LLMs on practical examples such as paratope prediction, humanization. However, due to the current size of the paper it will be carried out in a fully dedicated work for fine-tuning.
>
> - IgBleng is indeed a typo, thank you for pointing that out. For AbMPNN, we just wanted to use one model per category (one inverse-folding, and we took AntiFold simply because it was more recent and claimed better results). It is the reason that there is only AntiFold

---

### Meta-Review · Area_Chair_h2Cu · 2024-12-15

**Metareview:**

IgBlend integrates 3D structure and sequence data in an antibody LLM, showing improved performance in benchmarks and correlation with binding affinities.

Strengths include a well-written paper, effective pre-training framework, and strong empirical results.

Weaknesses include limited novelty, unfair comparisons, potential data leakage, and lack of code sharing. The benefits of the "Seq + Struct Guided" setting are unclear.

The decision to reject is influenced by concerns over novelty, unfair comparisons, and potential biases in the test set.

**Additional Comments On Reviewer Discussion:**

All the reviewers agreed that this paper is not ready for publishing.

---

### Decision · Program_Chairs · 2025-01-22

Reject